# A Closer Look to Positive-Unlabeled Learning from Fine-grained Perspectives: An Empirical Study

**Yuanchao Dai[1,2], Zhengzhang Hou[1,2], Changchun Li[1,2], Yuanbo Xu[1], En Wang[1], Ximing Li[1,2,3]***

[1]College of Computer Science and Technology, Jilin University, China
[2]Key Laboratory of Symbolic Computation and Knowledge Engineering, Jilin University, China
[3]RIKEN Center for Advanced Intelligence Project
{liximing86, yuanchaodai, changchunli93}@gmail.com

## Abstract

Positive-Unlabeled (PU) learning refers to a specific weakly-supervised learning paradigm that induces a binary classifier with a few positive labeled instances and massive unlabeled instances. To handle this task, the community has proposed dozens of PU learning methods with various techniques, demonstrating strong potential. In this paper, we conduct a comprehensive study to investigate the basic characteristics of current PU learning methods. We organize them into two fundamental families of PU learning, including *disambiguation-free empirical risks*, which approximate the expected risk of supervised learning, and *pseudo-labeling methods*, which estimate pseudo-labels for unlabeled instances. First, we make an empirical analysis on disambiguation-free empirical risks such as uPU, nnPU, and DistPU, and suggest a novel risk-consistent set-aware empirical risk from the perspective of aggregate supervision. Second, we make an empirical analysis of pseudo-labeling methods to evaluate the potential of pseudo-label estimation techniques and widely applied generic tricks in PU learning. Finally, based on those empirical findings, we propose a general framework of PU learning by integrating the set-aware empirical risk with pseudo-labeling. Compared with existing PU learning methods, the proposed framework can be a practical benchmark in PU learning.

## 1 Introduction

**P**ositive-**U**nlabeled (**PU**) learning refers to a specific weakly-supervised learning paradigm [1, 2, 3] for binary classification, which trains a binary classifier with a few positive labeled instances and massive unlabeled instances [4]. It arises in various practical scenarios such as automatic face tagging, spam detection, and Inlier-based outlier detection [5]. Due to its wide applicability, PU learning has increasingly attracted more attention from the machine learning community.

During the past decades, many emerging practical PU learning methods have been proposed with various advanced techniques [6, 7]. Because in PU learning, negative labeled instances are unavailable, how to deal with unlabeled instances becomes its key challenge; and from this taxonomic perspective, we organize the existing PU learning methods into two fundamental families, namely *disambiguation-free empirical risks* [5, 8, 9, 10] and *pseudo-labeling methods* [11, 12, 13, 14, 15, 16].

The disambiguation-free empirical risks, as the name suggests, directly apply only positive labeled instances and unlabeled instances to approximate the expected risk of supervised learning. Under certain data generation assumptions, previous studies suggest unbiased empirical risk uPU [17, 5] and several practical variants such as nnPU with non-negativity constraint [8], abs-PU with absolute-

---

*Corresponding author

value constraint [9], and DistPU with positive-class prior constraint [10]. In parallel, the basic idea of pseudo-labeling methods is estimating pseudo-labels for unlabeled instances, and training the binary classifier with them in a self-training manner. Analogous to semi-supervised learning, these methods typically estimate pseudo-labels by iteratively updating the current predictions. For example, RP [18] iteratively identifies reliable negative examples from unlabeled data and assigns hard pseudo-labels to them for subsequent training. Another recent Self-PU [11] utilizes a soft pseudo-labeling strategy that continuously refines label assignments by incorporating the evolving confidence scores throughout the training process. Additionally, they apply several generic tricks such as mixup augmentation, exponential moving average, and knowledge distillation to further improve the classification performance [7, 13, 19].

The current PU learning methods have demonstrated strong potential, but we find that most of them, especially pseudo-labeling ones, are commonly complicated by integrating with specific tricks. Accordingly, some of their basic characteristics are still unclear, such as what kind of pseudo-labeling techniques and generic tricks are practical. In this paper, we conduct a comprehensive study to investigate the basic characteristics from a fine-grained perspective of PU learning. First, we make an empirical analysis on disambiguation-free empirical risks; suggest a novel risk-consistent set-aware empirical risk from the perspective of aggregate supervision, and empirically validate that it can be a practical candidate for disambiguation-free empirical risks. Second, we turn to pseudo-labeling methods, and make an empirical analysis on basic techniques to estimate pseudo-label such as hard pseudo-labeling technique, soft pseudo-labeling technique, and high-confident pseudo-label selection strategies; additionally, we empirically analyze several widely applied generic tricks in PU learning. Finally, based on those empirical findings, we propose a general framework of PU learning by integrating the set-aware empirical risk with pseudo-labeling, namely **GPU**. We further suggest specification principles within GPU. Compared with existing PU learning methods, the proposed GPU framework can be a practical benchmark in PU learning. In summary, the contributions of this paper is outlined below:

- We conduct a comprehensive empirical study to the current PU learning methods, and make extensive empirical observations on the effectiveness of basic techniques and tricks in PU Learning.

- We propose a novel risk-consistent set-aware empirical risk from the perspective of aggregate supervision, which can be a practical candidate for disambiguation-free empirical risks, and then formulate a novel general framework of PU learning by integrating it with pseudo-labeling.

- We suggest implementation principles of GPU. Compared with existing PU learning methods, the proposed GPU framework can be a practical benchmark in PU learning.

## 2 Preliminaries

In this section, we review the problem setting of PU learning and the two main families of PU learning methods.

**Problem formulation and notations** Formally, under the two-sample problem setting [20] and completely selected at random (SCAR) assumption [21], given a positive dataset $\mathcal{D}_p = \{(\mathbf{x}_i, +1)\}_{i=1}^{n_p} \overset{i.i.d.}{\sim} p_p(\mathbf{x}) = p(\mathbf{x}|y = +1)$ with $n_p$ instances drawn from the positive-class conditional density $p_p(\mathbf{x})$ and an unlabeled dataset $\mathcal{D}_u = \{(\mathbf{x}_i, y_i)\}_{i=1}^{n_u} \overset{i.i.d.}{\sim} p(\mathbf{x})$ with $n_u$ instances drawn from the marginal density $p(\mathbf{x})$, where $\mathbf{x} \in \mathbb{R}^d$ and $y \in \{-1, +1\}$ denote the $d$-dimensional feature vector and the corresponding binary label, respectively. The objective of PU learning is to induce a classifier $g : \mathbb{R}^d \to \mathbb{R}$ over $\mathcal{D}_p \cup \mathcal{D}_u$, which can predict labels for unseen instances.

### 2.1 PU Learning with Disambiguation-free Empirical Risks

PU learning methods with disambiguation-free empirical risks directly utilize labeled positive data and unlabeled data to approximate the expected risk of supervised learning. In this work, we review several representative ones, including **uPU** [5], **nnPU** [8], **absPU** [9], and **DistPU** [10].

Table 1: A summary of basic techniques and widely applied generic tricks for PU learning methods with pseudo-labeling.

| PUL methods | pseudo-labeling technique | | | | generic trick | |
| | pseudo-label | | high confidence selection | | mixup | moving average |
| | hard | soft | with | w/o | | |
|---|---|---|---|---|---|---|
| RP [18] | ✓ | | | ✓ | | |
| AdaSampling [22] | ✓ | | | ✓ | | |
| GenPU [23] | ✓ | | | | | |
| Self-PU [11] | | ✓ | | ✓ | ✓ | |
| VPU [12] | | ✓ | ✓ | | ✓ | |
| PULNS [24] | ✓ | | | ✓ | | |
| P³Mix [13] | ✓ | | ✓ | | ✓ | |
| RobustPU [14] | ✓ | | | ✓ | | |
| HolisticPU [15] | ✓ | | ✓ | | | |
| LaGAM [25] | | ✓ | ✓ | | ✓ | ✓ |
| PUL-CPBF [16] | ✓ | | ✓ | | | |
| VQ-Encoder [26] | | ✓ | ✓ | | | |

Formally, let $\ell : \mathcal{X} \times \mathcal{Y} \to \mathbb{R}_+$ be any loss function, and $\pi = p(y = +1)$ be the positive-class prior. Since negative samples are not directly accessible in PU learning, the SCAR assumption fortunately provides a solution. Under this assumption, where the labeled positive examples are selected completely at random from all positive examples, given $R_p^+(g) = \mathbb{E}_{p_p(\mathbf{x})}\left[\ell\left(g(\mathbf{x}), +1\right)\right]$, $R_p^-(g) = \mathbb{E}_{p_p(\mathbf{x})}\left[\ell\left(g(\mathbf{x}), -1\right)\right]$, $R_n^-(g) = \mathbb{E}_{p_n(\mathbf{x})}\left[\ell\left(g(\mathbf{x}), -1\right)\right]$, and $R_u^-(g) = \mathbb{E}_{p(\mathbf{x})}\left[\ell\left(g(\mathbf{x}), -1\right)\right]$, we obtain that $p(\mathbf{x}) = \pi p_p(\mathbf{x}) + (1 - \pi)p_n(\mathbf{x})$, such that $(1 - \pi)R_n^-(g) = R_u^-(g) - \pi R_p^-(g)$. Then, suppose $\pi$ is known, one can formulate [5], which is an unbiased risk of PU learning uPU for the expected risk of supervised learning, formulated as follows:

$$R_{\text{uPU}}(g) = \pi R_p^+(g) + R_u^-(g) - \pi R_p^-(g), \tag{1}$$

and its empirical risk over $D_p \cup D_u$ is given below:

$$\widehat{R}_{\text{uPU}}(g) = \pi \widehat{R}_p^+(g) + \widehat{R}_u^-(g) - \pi \widehat{R}_p^-(g) \tag{2}$$

However, uPU suffers from overfitting when using flexible models due to negative empirical risk $\widehat{R}_u^-(g) - \pi \widehat{R}_p^-(g)$. Some methods attempt to impose a non-negative constraint on $\widehat{R}_u^-(g) - \pi \widehat{R}_p^-(g)$ to prevent the empirical risk from becoming negative. To achieve this, nnPU [8] incorporates the max function:

$$\widehat{R}_{\text{nnPU}}(g) = \pi \widehat{R}_p^+(g) + \max\{0, \widehat{R}_u^-(g) - \pi \widehat{R}_p^-(g)\}, \tag{3}$$

and absPU incorporates the absolute value function [9]:

$$\widehat{R}_{\text{absPU}}(g) = \pi \widehat{R}_p^+(g) + \left|\widehat{R}_u^-(g) - \pi \widehat{R}_p^-(g)\right| \tag{4}$$

Additionally, under the case of symmetric losses where $\ell(z) + \ell(-z) = 1$, we have $\widehat{R}_p^-(g) = 1 - \widehat{R}_p^+(g)$ naturally holds. Leveraging this property, Dist-PU [10] reformulates uPU as follows:

$$\widehat{R}_{\text{DistPU}}(g) = 2\pi \widehat{R}_p^+(g) + \left|\widehat{R}_u^-(g) - \pi\right|, \tag{5}$$

where the absolute value function is introduced to impose the non-negative constraint on $\widehat{R}_u^-(g) - \pi$.

## 2.2   PU Learning with Pseudo-labeling

PU learning methods with pseudo-labeling, as the name suggests, estimate pseudo-labels for unlabeled data, and train the classifier $g$ in a self-training manner [18, 22, 23, 24, 13, 14, 15, 16]. Referring to semi-supervised learning, we can formulate the generic objective of pseudo-labeling below:

$$\mathcal{L}(g; D_p, D_u) = \widehat{R}_p^+(g) + \mathcal{L}_u(g, \hat{y}; D_u), \tag{6}$$

where $\mathcal{L}_u(g, \hat{y}; D_u)$ is the self-training objective with unlabeled data, and $\hat{y}$ denotes the pseudo-label.

Table 2: Positive and negative label groups of datasets and the statistics of those PU learning sets.

| Dataset | $\pi$ | Positive Class | Negative Class | Feature | Train | Backbone |
|---|---|---|---|---|---|---|
| F-MNIST-1 | 0.4 | 0, 2, 4, 6 | 1, 3, 5, 7, 8, 9 | $28 \times 28$ | 60,000 | LeNet-5 |
| F-MNIST-2 | 0.6 | 1, 3, 5, 7, 8, 9 | 0, 2, 4, 6 | $28 \times 28$ | 60,000 | LeNet-5 |
| CIFAR-10-1 | 0.4 | 0, 1, 8, 9 | 2, 3, 4, 5, 6, 7 | $3 \times 32 \times 32$ | 50,000 | 7-Layer CNN |
| CIFAR-10-2 | 0.6 | 2, 3, 4, 5, 6, 7 | 0, 1, 8, 9 | $3 \times 32 \times 32$ | 50,000 | 7-Layer CNN |
| STL-10-1 | – | 0, 2, 3, 8, 9 | 1, 4, 5, 6, 7 | $3 \times 96 \times 96$ | 105,000 | 7-Layer CNN |
| STL-10-2 | – | 1, 4, 5, 6, 7 | 0, 2, 3, 8, 9 | $3 \times 96 \times 96$ | 105,000 | 7-Layer CNN |

We review existing pseudo-labeling methods and summarize the basic techniques and widely applied generic tricks in Table 1. Specifically, the basic problem of pseudo-labeling is the techniques to estimate pseudo-labels $\hat{y}$ for unlabeled data with current predictions, and they typically include **hard** and **soft** and pseudo-labeling techniques. Let $q = g(\mathbf{x})$ and $\phi(q) \in [0, 1]$ denote the prediction of the classifier and the confidence belonging to the positive class, respectively, where $\phi$ is a transformation function, and here we apply the sigmoid function. Then, the hard and soft pseudo-labels are estimated as $\hat{y} = \text{sign}(\phi(q) - 0.5) \in \{-1, +1\}$ and $\hat{y} = \phi(q)$, respectively. In addition, some studies suggest selecting high-confident pseudo-labels, rather than applying all of them [18, 22, 11, 24, 14], where the representatives include various thresholding strategies.

**Generic tricks** We briefly review two widely applied generic tricks in PU learning studies [11, 12, 13, 25], such as **mixup** and **moving average**. Mixup is an efficient data augmentation trick with the convex combination of instance pairs [7]. Given an instance pair $(\mathbf{x}_i, y_i)$ and $(\mathbf{x}_j, y_j)$, it generates an augmented instance $(\tilde{\mathbf{x}}, \tilde{y})$ as follows:

$$\tilde{\mathbf{x}} = \lambda \mathbf{x}_i + (1 - \lambda)\mathbf{x}_j, \quad \tilde{y} = \lambda y_i + (1 - \lambda)y_j, \qquad \lambda \sim \text{Beta}(\alpha, \alpha) \tag{7}$$

Here, the moving average refers to updating pseudo-labels with historical predictions during the classifier training process [27]. Formally, its update equation for pseudo-labels is given below:

$$\phi(q) \leftarrow \epsilon\phi(q^h) + (1 - \epsilon)\phi(q), \tag{8}$$

where $q^h$ denotes the historical prediction, and $\epsilon$ is a smoothing parameter.

## 3 Empirical Findings, Analysis, and Modifications

### 3.1 Settings of Empirical Study

We conduct empirical evaluations on 3 standard benchmark datasets, *i.e.* Fashion-MNIST (F-MNIST), CIFAR-10, and STL-10. Following [16], we transform them into a set of binary classification problems by partitioning their original 10 classes into positive and negative categories by varying the class prior $\pi \in \{0.4, 0.6\}$. For all datasets, the number of positive labeled instances is fixed as $n_p = 1,000$. The details of datasets are summarized in Table 2.

For each PU learning method, We employ dataset-appropriate backbones as follows: LeNet-5 for F-MNIST, 7-layer CNN for CIFAR-10 and STL-10; the MLP layer is used as the classification layer across all datasets. The mini-batch is fixed as 512 and the number of epochs is set to 100 for F-MNIST and 200 for others.

In addition, we employ the classification accuracy (**ACC**) as the evaluation metric. All experiments are conducted with five different random seeds on a server equipped with two Nvidia RTX4090 GPUs, and we report the mean and standard deviation of the results.

### 3.2 Disambiguation-free Empirical Risks

In this section, we suggest a novel set-aware empirical risk of PU learning and empirically evaluate it and existing disambiguation-free empirical risks with various surrogate loss functions.

Table 3: Properties of commonly used loss functions.

| Loss | Formula | Convex | Differ. | Symm. | Lipsc. |
|------|---------|--------|---------|-------|--------|
| hinge | $\max\{0, 1-z\}$ | ✓ | | | ✓ |
| logistic | $\log(1+e^{-z})$ | ✓ | ✓ | | ✓ |
| sigmoid | $1/(1+e^z)$ | | ✓ | ✓ | ✓ |
| squared | $(1-z)^2$ | ✓ | ✓ | | |
| ramp | $\min\{1, \max\{0, (1-z)/2\}\}$ | | | ✓ | ✓ |
| double-hinge | $\max\{0, (1-z)/2, -z\}$ | | | ✓ | ✓ |

**Set-aware empirical risk of PU learning**  In PU learning, we are given the positive labeled data and unlabeled data $\mathcal{D}_p \cup \mathcal{D}_u$, and the positive-class prior $\pi$. Inspired by previous weakly-supervised learning studies with aggregate supervision [28], we can arrange the training data as $\mathcal{D}_p \cup (\mathcal{D}_u, \pi)$, where we treat $(\mathcal{D}_u, \pi)$ as a set of instances with its approximate label proportion.[2] Accordingly, we can formulate the following set-aware empirical risk of PU learning (SAPU):

$$\widehat{R}_{\text{SAPU}}(g) = \widehat{R}_p^+(g) + \ell_{CE}\left(\frac{1}{n_u}\sum_{\mathbf{x}_i \in \mathcal{D}_u} g(\mathbf{x}_i), \pi\right), \tag{9}$$

where $\ell_{CE}$ denotes the cross-entropy loss. Because the size of $\mathcal{D}_u$ can be too large, directly fitting $\pi$ in the second term of $\widehat{R}_{\text{SAPU}}(g)$ may result in smoothing instance-level predictions. To alleviate this potential issue, we can randomly divide $\mathcal{D}_u$ into many subsets $\{\mathcal{S}_i\}_{i=1}^{n_s}$, where $\mathcal{S}_i = \{\mathbf{x}_{ij}\}_{j=1}^{S}$, $n_s$ is the number of subsets, and $S$ is the number of instances in each subset; and if $S$ is large enough, we can also approximate the label proportion of each subset as $\pi$. Upon these ideas, we can rearrange the training data as $\mathcal{D}_p \cup \{(\mathcal{S}_i, \pi)\}_{i=1}^{n_s}$, and then reformulate Eq.9 as follows:

$$\widehat{R}_{\text{SAPU}}(g) = \widehat{R}_p^+(g) + \frac{1}{n_s}\sum_{i=1}^{n_s}\ell_{CE}\left(\frac{1}{S}\sum_{\mathbf{x}_{ij} \in \mathcal{S}_i} g(\mathbf{x}_{ij}), \pi\right) \tag{10}$$

We consider SAPU as a practical candidate disambiguation-free empirical risks of PU learning. We show the following theorem to indicate that it is risk-consistent for the expected risk of supervised learning. The proof is presented in the Appendix.

**Lemma 3.1.** *Let $\hat{\pi}_i = \frac{1}{S}\sum_{i=1}^{S}\mathbf{1}[y_{ij} = +1]$ be the true proportion of positive instance in set $\mathcal{S}_i$. When the set size satisfies $S \geq \frac{3\pi(1-\pi)\log(2/\delta)}{2\epsilon^2}$, with probability at least $1-\delta$, we have $|\hat{\pi}_j - \pi| \leq \epsilon$ for each set $\mathcal{S}_i$ and $Var(\hat{\pi}_j) \leq \frac{\pi(1-\pi)}{S}$.*

**Theorem 3.2.** *Let $g^* = \arg\min_{g \in \mathcal{G}} R(g)$ is the minimizer of the true classification risk and $\hat{g}_{\text{SAPU}} = \arg\min_{g \in \mathcal{G}} \hat{R}_{\text{SAPU}}(g)$ denotes the minimizer of the risk form in Eq.10. Suppose that the pseudo-dimensions of $\{\mathbf{x} \mapsto g(\mathbf{x}) | g \in \mathcal{G}\}$ and $\{\mathbf{x} \mapsto \ell_{CE}(g(\mathbf{x}), \pi) | g \in \mathcal{G}\}$ are finite, and there exist constants $L_g, L_\ell$ such that $|g(\mathbf{x})| \leq L_g$ and $|\ell_{CE}(g(\mathbf{x}), \pi)| \leq L_\ell$ for all $\mathbf{x} \in \mathcal{X}$ and all $g \in \mathcal{G}$. Then, for any $\delta > 0$, with probability at least $1-\delta$:*

$$R(\hat{g}_{\text{SAPU}}) - R(g^*) \leq O\left(\sqrt{\frac{\log(1/\delta)}{n_p}}\right) + O\left(\sqrt{\frac{\log(1/\delta)}{n_s}}\right) + L_\ell \cdot O\left(\sqrt{\frac{\pi(1-\pi)\log(1/\delta)}{S}}\right) \tag{11}$$

**Results and analysis**  We empirically investigate the proposed SAPU and 4 existing disambiguation-free empirical risks with different commonly used loss functions. Table 3 presents 6 loss functions, *i.e.*, hinge, logistic, sigmoid, squared, ramp, and double-hinge, along with their mathematical formulations and theoretical properties. These loss functions are selected to represent diverse characteristics across 4 key properties: convexity, differentiability, symmetry, and Lipschitz continuity. Because the positive prior $\pi$ for the STL-10 dataset is unavailable, we conduct experiments on the CIFAR-10 and F-MNIST datasets.

---

[2]We declare that the label proportion approaches $\pi$, as $n_u$ goes to $\infty$

Table 4: The ACC scores (mean±std) of disambiguation-free empirical risks with widely used loss functions on F-MNIST and CIFAR-10. The highest scores are indicated in **bold**.

| Dataset | Method | $S$ | hinge | logistic | sigmoid | squared | ramp | double-hinge |
|---|---|---|---|---|---|---|---|---|
| F-MNIST-1 | uPU | - | 68.0±0.5 | 68.5±0.5 | 69.8±0.9 | **77.1±2.2** | 70.8±1.8 | 68.4±0.6 |
| | nnPU | - | 93.8±0.4 | 93.0±0.5 | 93.9±0.7 | 93.2±1.7 | 93.8±1.0 | **94.8±0.3** |
| | absPU | - | **94.2±0.4** | 93.3±0.5 | 93.3±0.7 | 93.7±0.4 | 93.6±0.6 | 94.1±0.2 |
| | Dist-PU | - | - | - | 94.3±0.4 | - | 94.0±0.2 | **94.7±0.2** |
| | SAPU | 32 | 93.9±1.0 | 95.9±0.2 | 95.9±0.2 | 94.6±0.6 | 94.5±1.0 | 94.5±0.9 |
| | | 64 | 92.8±0.3 | 96.0±0.2 | 96.0±0.1 | 93.0±0.0 | 93.8±0.8 | 94.0±0.9 |
| | | 128 | 92.8±0.1 | 96.1±0.0 | **96.2±0.1** | 91.0±0.5 | 93.5±0.8 | 92.9±0.1 |
| | | 256 | 93.0±0.5 | 96.0±0.2 | **96.2±0.0** | 90.8±0.1 | 92.9±0.5 | 92.8±0.3 |
| | | $n_u$ | 92.4±0.5 | 95.4±0.6 | 96.0±0.0 | 90.8±0.3 | 92.9±0.6 | 92.6±0.1 |
| F-MNIST-2 | uPU | - | 47.8±0.6 | 47.7±0.3 | 49.1±0.9 | **62.4±2.7** | 50.8±1.4 | 48.7±0.7 |
| | nnPU | - | 92.4±0.4 | 91.7±1.3 | 91.0±0.4 | 92.7±0.5 | 93.1±0.7 | **93.4±0.3** |
| | absPU | - | 92.6±0.6 | 91.5±0.7 | 91.0±1.2 | 92.0±0.3 | 92.4±0.7 | **93.4±0.4** |
| | Dist-PU | - | - | - | 91.3±0.9 | - | **93.3±0.6** | 92.2±0.3 |
| | SAPU | 32 | 94.5±0.5 | 95.7±0.0 | 95.8±0.2 | 94.7±0.5 | 95.3±0.0 | 94.9±0.3 |
| | | 64 | 94.6±0.1 | 95.8±0.0 | 95.8±0.1 | 92.8±0.5 | 94.0±0.4 | 94.5±0.5 |
| | | 128 | 93.6±0.4 | 95.9±0.2 | 96.0±0.1 | 90.8±0.2 | 93.2±0.4 | 93.9±0.3 |
| | | 256 | 93.7±0.2 | 95.8±0.2 | **96.1±0.0** | 88.2±1.8 | 93.4±0.5 | 93.2±0.0 |
| | | $n_u$ | 92.8±0.2 | 95.9±0.0 | 96.0±0.1 | 88.3±1.4 | 93.6±0.4 | 93.5±0.4 |
| CIFAR-10-1 | uPU | - | 80.5±0.7 | **81.7±0.9** | 81.6±1.9 | 66.1±2.4 | 77.3±2.3 | 79.9±0.7 |
| | nnPU | - | **86.4±0.4** | 84.3±0.7 | 85.1±1.4 | 83.2±1.0 | 86.1±0.5 | **86.4±0.1** |
| | absPU | - | 85.6±0.4 | 82.9±0.7 | 85.7±1.5 | 81.9±1.2 | **86.3±0.9** | 85.7±0.6 |
| | Dist-PU | - | - | - | 86.0±0.9 | - | 86.2±0.6 | **86.6±0.5** |
| | SAPU | 32 | 85.3±0.6 | 86.5±0.5 | 86.6±0.2 | 76.6±4.7 | 86.2±0.7 | 84.5±0.6 |
| | | 64 | 83.8±0.3 | 86.4±0.2 | 86.7±0.3 | 77.0±3.4 | 86.7±0.7 | 85.3±0.4 |
| | | 128 | 84.7±0.5 | 85.6±0.4 | **87.0±0.5** | 79.5±1.5 | 85.0±0.0 | 83.6±0.2 |
| | | 256 | 83.5±0.2 | 86.6±0.3 | 86.8±0.2 | 75.3±5.5 | 85.4±0.6 | 84.4±0.6 |
| | | $n_u$ | 84.1±0.2 | 85.3±0.4 | 86.8±0.3 | 78.7±2.5 | 85.5±1.2 | 84.1±0.7 |
| CIFAR-10-2 | uPU | - | 76.1±0.9 | **77.3±1.1** | 76.9±2.4 | 55.7±2.0 | 67.9±2.4 | 75.5±1.0 |
| | nnPU | - | **84.7±1.0** | 80.7±1.4 | 83.7±1.3 | 81.0±1.7 | 84.3±1.0 | 83.8±1.4 |
| | absPU | - | **84.4±0.9** | 78.0±1.7 | 83.8±1.4 | 79.3±2.7 | **84.4±0.7** | 82.4±1.4 |
| | Dist-PU | - | - | - | 82.1±1.1 | - | 83.4±1.6 | **85.6±0.7** |
| | SAPU | 32 | 60.7±0.0 | 85.1±0.7 | 85.1±0.7 | 81.3±0.3 | 74.7±6.3 | 74.7±6.3 |
| | | 64 | 62.4±1.7 | 85.2±0.7 | 84.9±0.7 | 81.3±0.1 | 74.9±6.1 | 75.7±5.3 |
| | | 128 | 61.6±0.9 | 85.2±0.7 | 84.7±0.6 | 81.1±0.0 | 72.3±9.1 | 72.6±8.7 |
| | | 256 | 60.7±0.0 | **85.4±0.6** | 84.5±0.5 | 81.3±0.0 | 73.8±7.4 | 74.0±7.1 |
| | | $n_u$ | 61.6±0.9 | 85.3±0.7 | 84.7±0.7 | 81.2±0.1 | 74.5±6.9 | 74.8±7.2 |

The experimental results in Table 4 demonstrate that the choice of loss function significantly influences classification accuracy across different datasets and methods. For F-MNIST dataset, the sigmoid loss consistently delivers superior performance, achieving a remarkable accuracy of 96% with our method. The double-hinge loss also performs exceptionally well, particularly with nnPU method and Dist-PU. On the more challenging CIFAR-10 dataset, the sigmoid loss still demonstrates robust performance (around 86% with SAPU), while the double-hinge loss excels in several configurations, notably with Dist-PU on CIFAR-10-2 (85.6%). Interestingly, the effectiveness of each loss function varies substantially across different empirical risk methods and datasets. For example, while SAPU achieves optimal results with sigmoid on CIFAR-10-1 (86.8%), its performance degrades considerably with the squared loss. Additionally, our empirical analysis confirms that smooth, differentiable losses (sigmoid, logistic) achieve better compatibility with SAPU's set-aware architecture, while non-smooth losses (double-hinge, ramp) align better with traditional point-wise optimization methods. This non-uniform behavior suggests a complex interaction between loss functions and model architectures

that cannot be reduced to simple heuristics, underscoring the importance of careful loss function selection based on specific application contexts.

Based on the above analysis, we can summarize the following guiding principles: (1) The smooth, differentiable losses (such as sigmoid, logistic loss) achieve better compatibility with the set-aware architecture of SAPU, while non-smooth losses (*e.g.*, double-hinge, ramp loss) align better with traditional point-wise optimization methods. (2) Convex losses generally provide better optimization guarantees. (3) Simple datasets (*e.g.*, F-MNIST) benefit from smooth losses, enabling fine-grained optimization, while complex datasets (*e.g.*, CIFAR-10) may require losses with stronger regularization properties.

Furthermore, as the core of SAPU lies in dividing unlabeled data into multiple subsets for set-aware supervision, we further conduct experiments to verify how subset size affects the performance of the model. We systematically tested different subset sizes $S = \{32, 64, 128, 256, n_u\}$ across all datasets and recorded classification accuracy with various loss functions. The experimental results in Table 4 demonstrate that $S = 256$ yields optimal performance on most datasets, particularly when combined with the sigmoid loss function. For example, the model achieved peak accuracies of 96.2% and 96.1% respectively on F-MNIST-1 and F-MNIST-2 when $S = 256$ with sigmoid loss function. For simpler datasets like F-MNIST, this phenomenon can be explained that when subsets are too small, individual subsets struggle to accurately reflect the overall label distribution; conversely, when subsets become excessively large (approaching $n_u$), instance-level predictions become overly smoothed, reducing the model's discriminative power. Moreover, for more complex datasets like CIFAR-10, our experiments indicate that larger subset sizes tend to be more effective. Based on our comprehensive analysis across different datasets, we recommend setting medium-sized subsets (e.g., $S = 256$) as a generally effective configuration for our SAPU method.

### 3.3 Pseudo-labeling Methods

In this section, we investigate the pseudo-labeling techniques and thresholding techniques for selecting high-confident pseudo-labels. For comprehensive evaluations, we first suggest several base methods and then discuss the empirical results.

**Base methods of pseudo-labeling**   We specify the generic objective of Eq.6 with specific pseudo-labeling techniques and thresholding strategies, leading to a set of base methods. First, we estimate pseudo-labels $\hat{y}$ by **hard** and **soft** pseudo-labeling techniques; and for clarity, we review them as $\hat{y} = \text{sign}(\phi(q) - 0.5) \in \{-1, +1\}$ and $\hat{y} = \phi(q)$, respectively. Second, the thresholding strategy refers to computing a threshold value $\tau$ to define the lower bound of high-confident pseudo-labels. Inspired by [29, 30], we specify 3 thresholding strategies to compute $\tau$, described below:

- **Fixed thresholding** treats the threshold value $\tau$ as a hyper-parameter, and empirically sets it as a constant value. Here, we fix $\tau$ to 0.95.

- **Adaptive thresholding** gradually updates the threshold value $\tau$ during classifier training. Following the idea that the predictions can be more accurate as the classifier continues to be trained [31], we gradually increase $\tau$ as follows:
  $$\tau \leftarrow \tau_{max} \times \min(1, t/T),$$
  where $t$ is the current epoch, $\tau_{max}$ is the maximum threshold value, and $T$ is the ramp-up period.

- **Class-specific adaptive thresholding** gradually updates the threshold values $\tau_p$ and $\tau_n$ for positive- and negative-classes, respectively. Following [30], we gradually update $\tau_p$ and $\tau_n$ as follows:
  $$\tau_p \leftarrow \tau_p \times \mathcal{C}_p^{(t)}, \qquad \tau_n \leftarrow \tau_n \times \mathcal{C}_n^{(t)},$$

where $\mathcal{C}_p^{(t)}$ and $\mathcal{C}_n^{(t)}$ are the ratios between the pseudo-label accuracies of positive- and negative-classes and their higher accuracy at epoch $t$.

Based on these specific techniques, we can specify **6 base methods of pseudo-labeling**.

**Results and analysis**   To comprehensively evaluate the effectiveness of different pseudo-label strategies and generic tricks under PU learning, we compare six base pseudo-labeling methods

Table 5: The ACC scores (mean±std) of 6 base methods of pseudo-labeling on F-MNIST and CIFAR-10. The highest scores are indicated in **bold**.

| Label | Threshold | M.A. | Mixup | F-MNIST-1 | F-MNIST-2 | CIFAR-10-1 | CIFAR-10-2 |
|---|---|---|---|---|---|---|---|
| Hard | Fixed | | | 90.0±1.7 | 92.7±0.1 | 85.1±0.6 | 83.1±3.8 |
| | | ✓ | | 89.9±1.6 | 89.3±2.3 | 84.2±1.8 | 82.0±3.5 |
| | | | ✓ | 90.5±0.8 | 92.8±0.2 | 85.1±0.3 | 85.0±1.0 |
| | | ✓ | ✓ | 89.6±1.9 | 89.0±1.7 | 84.0±2.1 | 82.1±3.7 |
| | Adaptive | | | 91.4±0.7 | 92.8±0.2 | 84.3±0.5 | 83.7±0.8 |
| | | ✓ | | 71.7±4.2 | 65.5±5.8 | 80.5±0.2 | 82.8±3.0 |
| | | | ✓ | 90.6±0.6 | 92.9±0.1 | 84.5±0.2 | 83.8±0.2 |
| | | ✓ | ✓ | 72.1±4.3 | 67.5±3.2 | 80.7±0.0 | 83.1±3.3 |
| | Class Adaptive | | | 91.5±0.4 | 89.0±0.0 | 84.3±0.4 | 83.5±0.5 |
| | | ✓ | | 71.7±4.2 | 65.5±5.8 | 82.0±1.0 | 82.0±1.5 |
| | | | ✓ | 91.0±1.0 | 89.6±0.2 | 84.5±0.5 | 84.0±0.5 |
| | | ✓ | ✓ | 72.1±4.3 | 65.5±5.7 | 80.5±1.0 | 82.5±1.5 |
| Soft | Fixed | | | 95.4±0.4 | 93.7±0.4 | 84.3±0.5 | 83.8±0.2 |
| | | ✓ | | 95.2±0.3 | 71.1±1.7 | 83.5±1.7 | 83.0±1.5 |
| | | | ✓ | 95.4±0.4 | 93.9±0.4 | 84.3±0.5 | 83.8±0.2 |
| | | ✓ | ✓ | 95.2±0.3 | 70.7±2.2 | 83.8±1.5 | 83.3±1.0 |
| | Adaptive | | | 95.4±0.4 | 94.1±0.3 | **85.9±0.5** | 82.9±3.4 |
| | | ✓ | | 74.0±4.0 | 69.9±3.0 | 83.5±1.5 | 83.0±1.3 |
| | | | ✓ | **95.5±0.4** | 94.2±0.4 | **85.9±0.5** | 84.8±0.0 |
| | | ✓ | ✓ | 82.9±2.4 | 70.7±2.2 | 83.9±1.3 | 83.4±1.1 |
| | Class Adaptive | | | 95.0±0.1 | 94.1±0.4 | 84.3±0.5 | 83.8±0.2 |
| | | ✓ | | 77.2±0.1 | 93.7±1.5 | 80.8±1.5 | 83.0±1.5 |
| | | | ✓ | **95.5±0.4** | **94.6±0.4** | 85.6±0.5 | **85.8±0.2** |
| | | ✓ | ✓ | 82.9±2.4 | 93.8±1.1 | 81.0±1.3 | 83.4±1.1 |

(including two pseudo-labeling techniques (hard vs. soft labeling) and three thresholding strategies (fixed, adaptive, and class-specific adaptive )) and two widely used enhancement techniques (mixup and moving average) on CIFAR-10 and F-MNIST datasets.

The results demonstrate that soft labeling consistently outperforms hard labeling, particularly when combined with class-adaptive thresholding on F-MNIST datasets (achieving up to 95.5%). Mixup proves to be the most consistent generic trick for ACC improvement across all experimental configurations, while moving average often leads to performance degradation when combined with other techniques. Mixup proves consistently beneficial because it addresses the fundamental challenge of decision boundary uncertainty in PU learning. By creating synthetic samples through convex combinations, mixup naturally smooths the decision boundaries in regions. This is particularly crucial in PU learning where the model must distinguish between true negatives and mislabeled positives within the unlabeled set. In contrast, the counterintuitive phenomenon of performance degradation with moving average techniques primarily stems from the unstable nature of pseudo-labels in PU learning. Unlike traditional semi-supervised learning where unlabeled data contains truly unlabeled instances, PU learning involves mislabeled negative samples, making historical predictions unreliable. The self-training process generates systematic biases, and moving average perpetuates rather than corrects these biases. On the other hand, moving average techniques may suppress the model's ability to rapidly adapt within the feature space to distinguish between positive and negative samples. Furthermore, the momentum parameter requires careful tuning, which significantly increases the experimental cost for hyperparameter optimization.

Overall, the combination of soft labeling with class-adaptive thresholds and Mixup yields the best performance across nearly all datasets. The only exception occurs in the CIFAR-10-1 dataset, likely due to its complex visual diversity as a natural image dataset, making the combination of fixed thresholding with moving average and mixup more suitable for handling its complex decision boundaries. These findings suggest that the combination of soft labeling, class-adaptive thresholding, and mixup generally constitutes the most promising universal method.

Table 6: The ACC scores (mean±std) of existing PU learning methods and GPU.

| Method | F-MNIST-1 | F-MNIST-2 | CIFAR-10-1 | CIFAR-10-2 | STL-10-1 | STL-10-2 | Rank |
|---|---|---|---|---|---|---|---|
| uPU | 77.1±2.2 | 62.4±2.7 | 81.7±0.9 | 77.3±1.1 | 76.7±0.8 | 71.5±4.8 | 15.3 |
| nnPU | 94.8±0.3 | 93.4±0.3 | 86.4±0.1 | 84.7±1.0 | 77.1±4.5 | 81.9±1.0 | 8.7 |
| absPU | 94.2±0.4 | 93.4±0.3 | 86.3±0.9 | 84.4±0.9 | 75.3±2.2 | 82.0±0.7 | 9.8 |
| Dist-PU | 94.7±0.2 | 93.3±0.6 | 86.7±0.5 | 85.6±0.7 | 78.3±0.8 | 81.5±1.1 | 8.7 |
| RP | 94.4±0.6 | 93.3±0.5 | 78.0±1.9 | 84.2±1.1 | 71.3±0.8 | 75.5±2.6 | 12.2 |
| AdaSampling | 93.6±0.3 | 93.5±0.2 | 79.6±0.5 | 79.1±1.0 | 74.3±2.2 | 82.6±0.8 | 11.3 |
| GenPU | 78.1±0.4 | 86.2±1.4 | 71.2±1.9 | 68.3±2.5 | 68.5±1.4 | 57.3±1.5 | 16.5 |
| Self-PU | 90.8±0.4 | 89.1±0.7 | 85.1±0.8 | 83.9±2.6 | 78.5±1.1 | 80.8±2.1 | 12.2 |
| VPU | 92.6±1.2 | 90.5±0.8 | 86.8±1.2 | 82.5±1.1 | 78.4±1.1 | 82.9±0.7 | 10.3 |
| PULNS | 91.0±0.5 | 89.1±0.8 | 87.2±0.6 | 83.7±2.9 | 80.2±0.8 | 83.6±0.7 | 9.8 |
| P³Mix-E | 92.6±0.4 | 91.8±0.2 | 88.2±0.4 | 84.7±0.5 | 80.2±0.9 | 83.7±0.7 | 7.8 |
| P³Mix-C | 92.8±0.6 | 90.4±0.1 | 88.7±0.4 | 87.9±0.5 | 80.7±0.7 | 84.1±0.3 | 6.5 |
| Robust-PU | 90.0±0.5 | 85.5±0.7 | 80.0±0.6 | 85.2±1.1 | 79.6±0.9 | 80.4±0.8 | 12.3 |
| HolisticPU | 96.2±0.1 | 96.0±0.3 | 91.0±0.3 | 90.4±0.5 | 82.5±0.5 | 84.0±1.2 | 3.2 |
| LaGAM | 94.9±0.2 | 94.1±0.3 | 89.9±0.3 | 88.0±1.4 | 85.3±0.3 | 85.0±0.3 | 2.8 |
| PUL-CPBF | 96.7±0.3 | 96.5±0.2 | 91.4±0.2 | 91.0±0.3 | 83.4±0.7 | 85.4±1.2 | 1.2 |
| **GPU** | 96.4±0.1 | 96.1±0.5 | 88.4±0.1 | 87.9±0.4 | 82.7±0.7 | 84.9±0.4 | 3.2 |
| PN learning | 97.7±0.1 | 97.7±0.1 | 91.9±0.1 | 91.9±0.1 | 86.0±0.6 | 86.0±0.6 | - |

## 3.4 Proposed GPU Framework

By integrating SAPU with pseudo-labeling, we suggest an efficient PU learning framework **GPU**. Its generic objective is given as follows:

$$\mathcal{L}_{\text{GPU}}(g) = \widehat{R}_p^+(g) + \mathcal{L}_u(g, \hat{y}; D_u) + \frac{\alpha}{n_s} \sum_{j=1}^{n_s} \ell_{CE}\left(\frac{1}{S} \sum_{\mathbf{x}_{ij} \in \mathcal{S}_i} g(\mathbf{x}_{ij}), \pi\right), \qquad (12)$$

where $\alpha$ is a coefficient parameter.

We can interpret GPU as a regularized pseudo-labeling method of PU learning, where the set-aware term is treated as a regularization term. Based on the previous evaluations, we find that pseudo-labeling methods depend on high quality of pseudo-labels in the early training stage because they are in a self-training manner. Accordingly, we suggest a **warm-up stage** by minimizing the objective of SAPU. In addition, we can specify the pseudo-labeling techniques and thresholding strategies according to our empirical observations.

**Results and analysis** To evaluate the efficacy of our proposed GPU framework compared to existing PU learning methods, we conduct experiments on F-MNIST, CIFAR-10, and STL-10 datasets to assess its general performance across varying scenarios. For comprehensive comparison, we also include PN learning (*i.e.*, supervised learning) as an upper bound baseline. Our GPU implementation uses a subset size $S = 256$ with the sigmoid loss function and employs soft pseudo-labeling with class-adaptive thresholding based on our empirical observations. For the warm-up stage, we train using only SAPU for 20 epochs before introducing the pseudo-labeling component.

As evident from Table 6, our proposed GPU framework demonstrates competitive performance across all benchmark datasets, ranking 3.2 overall, tied with HolisticPU and slightly behind LaGAM (2.8), while remaining competitive with the leading PUL-CPBF. For example, GPU achieves accuracy scores of 96.4% and 96.1% on F-MNIST-1 and F-MNIST-2 respectively, which are comparable to the best-performing PUL-CPBF (96.7% and 96.5%). For CIFAR-10, GPU obtains 88.4% and 87.6%, positioning it among the top-tier methods but slightly below HolisticPU and PUL-CPBF. Similar competitive performance is also demonstrated on the STL-10 dataset. The performance gap between GPU and the best-performing methods reflects our focus on exploring fundamental techniques and integrating them with our novel set-aware empirical risk SAPU, rather than employing sophisticated techniques like ensemble methods in PUL-CPBF, trend detection in HolisticPU, or meta-learning in

LaGAM. GPU provides a general framework that can integrate future advances, while specialized methods may not generalize well. Notably, GPU significantly outperforms traditional PU learning methods across all datasets, demonstrating that combining set-aware empirical risk estimation with pseudo-labeling strategies effectively enhances the discriminative capability of the model.

## 4 Discussion and Future Works

In this paper, we comprehensively review the current families of PU learning and investigate their basic characteristics. We review the existing disambiguation-free empirical risks and suggest a novel set-aware empirical risk SAPU from the perspective of aggregate supervision, which is risk-consistent for the expected risk of supervised learning. We empirically evaluate them with various commonly applied loss functions. In addition, we review the basic techniques and widely applied generic tricks, *i.e.* mixup and moving average, in the existing pseudo-labeling methods. To empirically evaluate them, we formulate a set of base methods specified by hard and soft pseudo-labeling techniques with thresholding strategies for selecting high-confident pseudo-labels such as fixed, adaptive, and class-specific adaptive thresholding strategies. Finally, we propose an efficient PU learning framework GPU by integrating SAPU with pseudo-labeling. GPU involves a warm-up stage by minimizing SAPU and specify the framework according to our empirical observations. We compare GPU with the existing PU learning methods, and the empirical results demonstrate that GPU can be a practical benchmark in PU learning, and is scalable for future pseudo-labeling techniques.

In the future, there are two potential problems that require more attention. One basic problem is how to estimate more accurate pseudo-labels [32, 33, 34] since we only investigate the straightforward pseudo-labeling techniques. Some advanced techniques such as ensemble learning [35] demonstrate strong potential. Another problem is whether the existing PU learning can be effective for the scenarios with scarce positive labeled instances and how to deal with such scenarios, which can appear in many real-world applications.

## Acknowledgements

We would like to acknowledge support for this project from the National Science and Technology Major Project (No.2021ZD0112500), and the National Natural Science Foundation of China (No.62276113).

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

# A Proof of Lemma 3.1

**The boundary of the bag deviation** Under the SCAR assumption [21], each sample in the unlabeled dataset has an independent probability $\pi$ of being positive. Given a bag $j$ containing $s$ samples, since the variance of a Bernoulli random variable is $\pi(1-\pi)$, we can obtain a tighter bound using Bernstein's inequality [36] for any $\epsilon > 0$:

$$P(|\hat{\pi}_j - \pi| \geq \epsilon) \leq 2\exp\left(-\frac{S\epsilon^2}{2\pi(1-\pi) + 2\epsilon/3}\right) \leq \delta \tag{13}$$

Then,

$$S \geq \frac{3\pi(1-\pi)}{2\epsilon^2}\log(2/\delta) \tag{14}$$

**The variance of $\hat{\pi}_j$** The variance of $\hat{\pi}_j$ can be given by the mean of $S$ independent Bernoulli random variables:

$$\mathrm{Var}(\hat{\pi}_j) = \frac{\pi(1-\pi)}{S} \tag{15}$$

# B Proof of Theorem 3.2

We decompose the excess risk as:

$$R(\hat{g}_{\mathrm{SAPU}}) - R(g^*) \leq$$
$$\underbrace{|R(\hat{g}_{\mathrm{SAPU}}) - R_{\mathrm{SAPU}}(\hat{g}_{\mathrm{SAPU}})|}_{\text{Term 1}} + \underbrace{|R_{\mathrm{SAPU}}(\hat{g}_{\mathrm{SAPU}}) - R_{\mathrm{SAPU}}(g^*)|}_{\leq 0} + \underbrace{|R_{\mathrm{SAPU}}(g^*) - R(g^*)|}_{\text{Term 2}} \tag{16}$$

For Term 1, using the uniform convergence theory and the fact that the deviation in bag proportions is bounded by $\epsilon$, we have:

$$|R(\hat{g}_{\mathrm{SAPU}}) - R_{\mathrm{SAPU}}(\hat{g}_{\mathrm{bag}})| \leq C_1\sqrt{\frac{d\log(n_p) + \log(1/\delta)}{n_p}} + C_2\sqrt{\frac{d\log(n_s) + \log(1/\delta)}{n_s}} + \epsilon L_\ell \tag{17}$$

where $L_\ell$ is the Lipschitz constant of the cross-entropy loss; $C_1$ and $C_2$ are universal constants; $d$ is the pseudo-dimension of the function class.

According to Hoeffding's inequality, for any $\delta > 0$, $|\hat{\pi}_j - \pi| \leq \sqrt{\frac{\log(2/\delta)}{2S}}$ holds with probability at least $1 - \delta$. Then, we have:

$$|R(\hat{g}_{\mathrm{SAPU}}) - R_{\mathrm{SAPU}}(\hat{g}_{\mathrm{bag}})| \leq C_1\sqrt{\frac{d\log(n_p) + \log(1/\delta)}{n_p}} + C_2\sqrt{\frac{d\log(n_s) + \log(1/\delta)}{n_s}} + L_\ell\sqrt{\frac{\log(2/\delta)}{2S}} \tag{18}$$

For Term 2, the deviation comes from the difference between the true positive class prior $\pi$ and the bag proportions $\hat{\pi}_j$. Using the variance bound from Lemma 3.1 and applying Jensen's inequality:

$$|R_{\mathrm{SAPU}}(g^*) - R(g^*)| \leq L_\ell\sqrt{\mathbb{E}[(\hat{\pi}_j - \pi)^2]} = L_\ell\sqrt{\mathrm{Var}(\hat{\pi}_j)} = L_\ell\sqrt{\frac{\pi(1-\pi)}{S}} \tag{19}$$

Then,

$$R(\hat{g}_{\text{\tiny SAPU}}) - R(g^*) \leq C_1 \sqrt{\frac{d \log(n_p) + \log(1/\delta)}{n_p}} + C_2 \sqrt{\frac{d \log(n_s) + \log(1/\delta)}{n_s}}$$

$$+ L_\ell \sqrt{\frac{\log(2/\delta)}{2S}} + L_\ell \sqrt{\frac{\pi(1-\pi)}{S}}$$

$$= C_1 \sqrt{\frac{d \log(n_p) + \log(1/\delta)}{n_p}} + C_2 \sqrt{\frac{d \log(n_s) + \log(1/\delta)}{n_s}}$$

$$+ L_\ell \left( \sqrt{\frac{\log(2/\delta)}{2S}} + \sqrt{\frac{\pi(1-\pi)}{S}} \right)$$

$$= O\left( \sqrt{\frac{\log(1/\delta)}{n_p}} \right) + O\left( \sqrt{\frac{\log(1/\delta)}{n_s}} \right) + L_\ell \cdot O\left( \sqrt{\frac{\pi(1-\pi)\log(1/\delta)}{S}} \right)$$

$$(20)$$

## C   Limitations and Broader Impacts

### C.1   Limitations

Despite our comprehensive empirical study, accurately estimating pseudo-labels remains challenging, especially with limited positive samples. Our methods could be further improved by incorporating advanced techniques such as ensemble learning to generate more reliable pseudo-labels. Additionally, the set-aware empirical risk method may face challenges with extremely imbalanced datasets where the positive class prior becomes difficult to estimate accurately.

### C.2   Broader Impacts

Our GPU framework introduces a novel perspective by integrating empirical risk with pseudo-labeling methods, enhancing PU learning applicability in real-world scenarios such as medical diagnoses and fraud detection. The proposed set-aware empirical risk extends the theoretical foundation of PU learning through aggregate supervision, which could inspire further weakly-supervised learning research. By making PU learning more reliable with limited labeled data, our work contributes to reduced annotation costs and broader accessibility of machine learning in resource-constrained environments.

