# OpenReview forum: "A Closer Look to Positive-Unlabeled Learning from Fine-grained Perspectives: An Empirical Study"
_NeurIPS.cc/2025/Conference — NeurIPS 2025 poster_

### Official Review · Reviewer_REBb · 2025-06-02

**Clarity:** 3
**Significance:** 1
**Originality:** 3
**Rating:** 2
**Confidence:** 5

**Summary:**

This paper investigates a classic problem: PU Learning. It analyzes and compares two common types of algorithms for this problem (based on ERM and Pseudo-label), and proposes a new algorithm (GPU) by integrating the two categories based on empirical studies. The proposed method is evaluated through experiments on three typical datasets.

**Questions:**

**1. Problem**

In real-world applications, one of the key challenges in PU Learning is the unknown positive sample distribution $\pi$. Although techniques like MPE [1]/BBE [2] can estimate it, their accuracy in high-dimensional data remains uncertain. Could the author please discuss whether this issue was considered in the research?

**2.Related Work**

While the review of PU Learning is comprehensive, several representative studies have not been discussed or compared. For example: [2, 3, 4]. Could the author please include a discussion on how these works relate to the research? Comparing the method with these studies would help clarify the unique contributions and positioning of the approach within the field.

**3. Method**
* What is the specific motivation for combining SAPU and pseudo-labeling in the method? Which particular challenges in PU Learning does this combination address?
* Similar algorithmic designs (ERM && pseudo-labeling) exist commonly in other weakly supervised learning problems [5] (e.g., UU Learning [6], CL Learning [7, 8], Learning with New Class [9, 10]). Would the method be applicable to these scenarios as well? Elaborating on the generalizability of the approach across related learning problems would add depth to the discussion.

**4. Experiments**

To enhance the reproducibility and interpretability of the results, could the author please provide the following details:
* Why were different backbones (LeNet-5 && CNN) used for different datasets instead of a unified one?
* What was the rationale behind setting different epoch numbers for each dataset?
* How were the positive and negative categories defined for each dataset? Different classification methods can lead to drastically different problem difficulties, as the difficulty of distinguishing between some classes is low while that of others is high.
* Could the author elaborate on the process of setting and tuning hyperparameters (e.g., learning rates, weight decay)?

Providing these details would greatly assist readers in understanding the experimental design and reproducing the results, thereby strengthening the credibility of the findings.

[1] Adjusting the outputs of a classifier to new a priori probabilities: a simple procedure

[2] Mixture proportion estimation and PU learning: A modern approach

[3] Positive-unlabeled learning with label distribution alignment

[4] Positive-unlabeled learning using random forests via recursive greedy risk minimization

[5] Machine Learning from Weak Supervision: An Empirical Risk Minimization Approach

[6] On the minimal supervision for training any binary classifier from only unlabeled data

[7] Learning from complementary labels

[8] Unbiased risk estimators can mislead: A case study of learning with complementary labels.

[9] Handling New Class in Online Label Shift

[10] An Unbiased Risk Estimator for Learning with Augmented Classes

**Ethical Concerns:**

["NO or VERY MINOR ethics concerns only"]

**Final Justification:**

In summary, while the paper provides a comprehensive empirical study and practical implementation guidelines, its core contributions lack the originality and depth required for acceptance at this venue. I encourage the authors to refine their research focus and explore more impactful innovations in future work.

**Limitations:**

yes

**Quality:**

2

**Strengths And Weaknesses:**

**Strengths**

* The paper is well-structured and written in a fluent manner, making it easy to follow and understand.
* It provides a comprehensive review of PU Learning, demonstrating a solid grasp of the research landscape in this field.

**Weaknesses**

The core research objective of this work remains ambiguous:
* If the study aims to establish a benchmark, expanding the experimental scope with more diverse datasets, detailed parameter configurations (e.g., learning rates) would significantly enhance reproducibility;
* If the study intends to propose a novel algorithm, the performance needs substantial enhancement, as the method currently fails to outperform state-of-the-art approaches under most experimental settings.

Addressing this ambiguity and reinforcing either the benchmark rigor or algorithmic superiority would strengthen the paper's scientific contribution.

---

> ### Author Rebuttal · Authors · 2025-07-31
>
> **Q1. Could the author discuss whether the issue of estimating the positive sample distribution was considered in the research?**
>
> Thank you for your comment. While unknown positive class prior is indeed an important consideration in real-world applications, we would like to clarify our research positioning. Our work focuses on investigating the fundamental characteristics and techniques within existing PU learning paradigms, rather than addressing the prior estimation problem specifically. The assumption of known positive class prior is widely adopted in the PU learning literature.
>
> Our work focuses on investigating the fundamental characteristics and techniques within existing PU learning paradigms, rather than addressing the prior estimation problem specifically. The assumption of known π is widely adopted in recent PU learning methods we compare against, as it allows for focused investigation of other critical aspects such as empirical risk design and pseudo-labeling strategies. Additionally, our proposed SAPU framework is orthogonal to prior estimation techniques, and can readily incorporate prior estimation methods like MPE/BBE as a preprocessing step. We believe combining SAPU with robust prior estimation represents a valuable direction for future work.
>
>
> &nbsp;
>
>
> **Q2. Several representative studies have not been discussed or compared. Could the author include a discussion on how these works relate to the research?**
>
> Thank you for your suggestion. [2] addresses the fundamental challenge of estimating the positive class prior $\pi$. As discussed in our response to the first question, this work is orthogonal to our research, since our method assumes $\pi$ is known to focus on empirical risk design and pseudo-labeling strategies.
>
> [3] represents an extension of Dist-PU that we discussed in our disambiguation-free empirical risks family. [3] addresses the challenge of distribution mismatch between the positive labeled data and the positive instances within unlabeled data, which can violate the SCAR assumption. The label distribution alignment technique provides a way to handle such distribution shifts, making it a valuable enhancement to the Dist-PU.
>
> [4] presents an alternative algorithmic method using random forests rather than neural networks we primarily focus on.
> We will expand the related work and discuss their relationship to our proposed method in the next version.
>
>
> &nbsp;
>
>
> **Q3. What is the specific motivation for combining SAPU and pseudo-labeling in the method? Which particular challenges in PU Learning does this combination address?**
>
> Thank you for your questions. The motivation for combining SAPU and pseudo-labeling lies in addressing the complementary limitations of existing disambiguation-free empirical risks and pseudo-labeling methods in PU learning. Our empirical analysis revealed that disambiguation-free empirical risks, while theoretically grounded, may suffer from gradient instability, particularly when the negative empirical risk $ R_u^-(g) - \pi R_p^-(g) $ becomes negative. Conversely, pseudo-labeling methods can capture complex instance-level patterns but are highly dependent on pseudo-label quality, especially in early training stages where predictions are unreliable. By combining SAPU with pseudo-labeling into our GPU framework, we address both challenges simultaneously: SAPU provides stable set-level supervision through aggregate constraints that regularize the learning process, while pseudo-labeling enables fine-grained instance-level discrimination. This combination creates a more robust training objective.
>
>
> &nbsp;
>
>
> **Q4. Elaborating on the generalizability of the approach across related learning problems (e.g., UU Learning, CL Learning, Learning with New Class).**
>
> Thank you for this insightful question. The core innovation of our SAPU method lies in leveraging aggregate supervision through set-level label proportions, which naturally extends to various weakly supervised learning paradigms where similar distributional information is available. For instance, in UU learning, where we have two unlabeled datasets from different distributions, SAPU can be adapted by treating each unlabeled set with its estimated class proportion, formulating a cross-entropy loss that encourages the model to match these proportions at the set level. The mathematical foundation remains consistent: instead of $\pi$ representing the positive class prior in PU learning, we would have $\pi_1$ and $\pi_2$ representing the class distributions in the two unlabeled datasets.
>
> For learning with new class, our GPU framework becomes particularly relevant. The pseudo-labeling component can be adapted to handle the discovery and classification of new classes by incorporating uncertainty estimation and cluster-based pseudo-labeling strategies. The set-aware supervision from SAPU can help maintain class proportion constraints even when new classes emerge.
>
>
> &nbsp;
>
>
> **Q5. Experimental design choices (backbones, epochs, and class definitions).**
>
> Thank you for your comment. We want to clarify that our experimental setup follows well-established practices in the PU learning community [1], which we cited and followed for consistency and fair comparison. The different epoch settings reflect established convergence patterns observed across the PU learning community for these standard benchmark datasets. F-MNIST typically reaches convergence faster due to its simpler feature space and lower intra-class variation, while CIFAR-10 and STL-10 require longer training periods to adequately learn the more complex decision boundaries present in natural images. These settings align with the training schedules used in the reference methods we compare against, ensuring that our experimental conditions provide fair and meaningful comparisons.
>
> [1] Positive and unlabeled learning with controlled probability boundary fence.
>
>
> &nbsp;
>
>
> **Q6. Could the author elaborate on the process of setting and tuning hyperparameters (e.g., learning rates, weight decay)?**
>
> Thank you for your comment. We set learning rates as $5e-3$ for F-MNIST and $1e-4$ for others, and set weight decay as $1e-6$. We will add them in the next version.

---

> > ### Comment · Reviewer_REBb · 2025-08-03
> >
> > Thank you for your thoughtful reply and the additional clarifications provided! I appreciate the effort to address some of the concerns raised in the initial review. While your responses have partially resolved certain questions, the current experimental results remain insufficient to fully substantiate the claimed contributions of this work. This aligns with my primary concern highlighted in the initial review (see Weakness section). Given the methodological novelty and depth required for acceptance at NeurIPS, I retain the original score.
> >
> > 1. **Limitations in Experimental Validation.** I acknowledge the empirical analysis conducted on PU learning methods. However, as highlighted by *Reviewer Bw6s*, the datasets used (e.g., F-MNIST, CIFAR-10, STL-10) are relatively small and simplistic, which limits the ability to validate the proposed framework’s scalability and robustness. Notably, the proposed algorithm underperforms existing methods in several critical settings. For example, on CIFAR-10-2 with hinge loss, the SAPU method achieves only 60% accuracy (Table 4), significantly lagging behind the ERM baseline (84%). This raises concerns about whether the method is inherently dependent on specific loss functions for effectiveness. Even in the final comparison (Table 6), the algorithm demonstrates only marginal improvements over strong baselines, such as 87.9% vs. 91.0% on CIFAR-10-2.
> >
> > 2. **Insufficient Methodological Novelty.** As also mentioned by *Reviewer Bw6s*, while the integration of SAPU with pseudo-labeling is an interesting direction, the methodological novelty remains limited. The proposed GPU framework recombines existing components (e.g., risk estimation and soft pseudo-labeling) without introducing fundamentally new theoretical insights or architectural innovations. Furthermore, techniques such as mixup and moving average—which are well-established in the broader machine learning literature (e.g., the seminal mixup paper [4] has been cited over 13,000 times)—are employed here without demonstrating novel adaptations or clear advantages specific to the PU learning context.
> >
> > In summary, while the paper provides a comprehensive empirical study and practical implementation guidelines, its core contributions lack the originality and depth required for acceptance at this venue. I encourage the authors to refine their research focus and explore more impactful innovations in future work.

---

> > > ### Comment · Reviewer_REBb · 2025-08-07
> > >
> > > In light of no response received from the authors, I will maintain my original score. While the work presents interesting ideas, I believe the experimental results are not sufficiently compelling for acceptance at NeurIPS. I remain open to further discussion should the authors wish to engage.

---

> > > > ### Author Response · Authors · 2025-08-09
> > > > **Overall Response**
> > > >
> > > > &nbsp;&nbsp; Thank you for your continued engagement and valuable feedback. We appreciate the opportunity to clarify the positioning and contributions of our work.
> > > >
> > > > &nbsp;&nbsp; Our work is primarily positioned as **an empirical study** that conducts systematic analysis of existing PU learning methods from fine-grained perspectives. Through detailed evaluation of disambiguation-free empirical risks and pseudo-labeling methods across different loss functions, thresholding strategies, and generic tricks, we reveal several previously unknown patterns in PU learning. For instance, we discover that sigmoid loss consistently outperforms others with set-aware supervision (achieving 96% on F-MNIST), while hinge loss shows poor compatibility, and that soft labeling combined with class-adaptive thresholding and mixup consistently outperforms other combinations. Moving average, contrary to its behavior in semi-supervised learning, often degrades performance in PU learning contexts. These empirical findings provide evidence-based guidelines for technique selection that we believe will be valuable for the PU learning community.
> > > >
> > > > &nbsp;&nbsp; On the other hand, our work also introduces an aggregate supervision perspective for understanding PU learning, which we believe offers a fresh theoretical viewpoint. Through our proposed SAPU method, we demonstrate how PU learning can be connected with broader weakly supervised learning theory, potentially enabling future research to explore cross-domain technique transfer between different weakly supervised learning problems such as UU learning, complementary learning, and learning with new classes. While our theoretical contribution may be incremental, we hope this perspective opens up new possibilities for unified analysis across multiple weakly supervised learning domains and provides a foundation that future work can build upon to develop more sophisticated methods.
> > > >
> > > > &nbsp;&nbsp; We sincerely appreciate the reviewer's thoughtful and constructive feedback, which helps us better articulate the positioning and value of our work. We believe our empirical findings provide useful guidance for technique selection in PU learning, while our aggregate supervision perspective offers a different lens for understanding these problems. We hope that our analysis and insights can contribute to the ongoing development of this important area.

---

### Official Review · Reviewer_Bw6s · 2025-06-30

**Clarity:** 3
**Significance:** 3
**Originality:** 2
**Rating:** 4
**Confidence:** 3

**Summary:**

This paper categorizes existing PU learning methods into two types and summarizes or improves them, respectively. For the first type of methods, this paper proposes SPU based on aggregate supervision and proves that it is risk-consistent. It also achieves good performance. For the second type of methods, this paper summarizes their techniques and evaluates the tricks used in these methods. Finally, this paper combines these two types of methods and proposes GPU, which demonstrates excellent performance on multiple datasets.

**Questions:**

1. Why does the choice of loss function in Table 4 have such a significant impact? How should we choose an appropriate loss function?

**Ethical Concerns:**

["NO or VERY MINOR ethics concerns only"]

**Final Justification:**

The detailed and thoughtful rebuttal has addressed most of my concerns. My concern about novelty, which is my major concern, is well replied by the authors. The lack of introduction to aggregate supervision is promised to be handled in the revised version. But the choice of dataset still need to be improved. Thus, I decided to keep my score.

**Limitations:**

Yes.

**Paper Formatting Concerns:**

There are no major formatting issues in this paper.

**Quality:**

2

**Strengths And Weaknesses:**

Strengths:

1. This article provides a thorough discussion of many existing works and offers many valuable tricks.
2. The performance of GPU in most scenarios is promising.
3. The article is clearly written.

Weaknesses:

1. The novelty is limited. SAPU is simply a straightforward application of aggregate supervision in PU learning.
2. This article seems to be merely a summary of existing methods. Although it indicates that certain tricks are effective, it does not explain why.
3. This article lacks a detailed introduction to aggregate supervision.
4. The datasets used in this paper are too small and too simple. The authors should conduct experiments on larger and more realistic datasets.

---

> ### Author Rebuttal · Authors · 2025-07-31
>
> **Q1. SAPU is simply a straightforward application of aggregate supervision in PUL.**
>
> Thank you for this comment. While our SAPU is inspired by aggregate supervision, we argue that this is not a straightforward application but represents a principled innovation based on risk estimation theory. First, SAPU is built upon the theoretical foundation of disambiguation-free empirical risks, where we innovate in the risk estimation term by replacing the estimated negative risk term $\widehat{R}\_u^-(g) - \pi \widehat{R}\_p^-(g)$ (which causes negative risk issues in flexible models) with a theoretically grounded set-aware cross-entropy term that avoids optimization instabilities. Second, our subset partitioning strategy with formal variance analysis provides theoretical guarantees on proportion estimation accuracy, which is showed in Lemma 3.1 of our paper. More importantly, this work aims to establish a practical benchmark for PU learning by providing comprehensive empirical analysis and a unified framework. The integration framework GPU demonstrates how to effectively combine risk estimation with pseudo-labeling through careful design principles. So the novelty of our work lies not just in individual components but in establishing methodological principles that can guide future PU learning research.
>
>
> &nbsp;
>
>
> **Q2. Although it indicates that certain tricks are effective, it does not explain why.**
>
> Thank you for your suggestions.
>
> **Mixup** proves consistently beneficial because it addresses the fundamental challenge of decision boundary uncertainty in PU learning. By creating synthetic samples through convex combinations, mixup naturally smooths the decision boundaries in regions. This is particularly crucial in PU learning where the model must distinguish between true negatives and mislabeled positives within the unlabeled set.
>
> The counterintuitive phenomenon of performance degradation with **moving average** techniques primarily stems from the unstable nature of pseudo-labels in PU learning. Unlike traditional semi-supervised learning where unlabeled data contains truly unlabeled instances, PU learning involves mislabeled negative samples, making historical predictions unreliable. The self-training process generates systematic biases, and moving average perpetuates rather than corrects these biases. On the other hand, moving average techniques may suppress the model's ability to rapidly adapt within the feature space to distinguish between positive and negative samples. Furthermore, the momentum parameter requires careful tuning, which significantly increases the experimental cost for hyperparameter optimization.
>
>
> &nbsp;
>
>
> **Q3. This article lacks a detailed introduction to aggregate supervision.**
>
> Thank you for your comment. Aggregate supervision refers to learning from group-level constraints rather than instance-level labels. In traditional supervised learning, we have individual labels $y_i$ for each instance $x_i$. In aggregate supervision, we only know aggregate statistics about groups of instances, such as the proportion of positive instances in a set. Our method leverages this principle by treating the unlabeled dataset $D_u$ with class prior $\pi$ as aggregate supervision. Instead of trying to predict individual labels, we divide $D_u$ into subsets $ \\{\mathcal{S}_i\\}\_{i=1}^S$ and enforce that each subset's predicted positive proportion approximates $\pi$. This strategy avoids the challenging individual pseudo-labeling problem and provides a natural regularization for overfitting. We will add the explanation about aggregate supervision in the next version.
>
>
> &nbsp;
>
>
> **Q4. Why does the choice of loss function in Table 4 have such a significant impact? How should we choose an appropriate loss function?**
>
> Thank you for these comments. The significant impact of loss function on the performance of SAPU reflects a fundamental mathematical incompatibility rather than algorithmic deficiency. Our SAPU employs cross-entropy loss for the set-level supervision term (Eq.9-10), which expects continuous probability outputs and relies on smooth gradient flow for effective optimization. In contrast, double-hinge loss is piecewise linear and promotes discrete, margin-based decisions through its sharp transitions at $z=1$ and $z=0$. This creates conflicting optimization dynamics where the set-aware component pulls toward smooth probabilistic outputs while the instance-level double-hinge component pushes toward sharp margin-based decisions. It explains why SAPU shows sensitivity to loss function compared to traditional disambiguation-free empirical risks (*e.g.,*  uPU, nnPU).
>
> Based on the above analysis, we can summarize the following guiding principles:
>
> (1) The smooth, differentiable losses (such as sigmoid, logistic loss) achieve better compatibility with the set-aware architecture of SAPU, while non-smooth losses (such as double-hinge, ramp loss) align better with traditional point-wise optimization methods.
>
> (2) Convex losses generally provide better optimization guarantees.
>
> (3) Simple datasets (*e.g.,* F-MNIST) benefit from smooth losses, enabling fine-grained optimization, while complex datasets (*e.g.,* CIFAR-10) may require losses with stronger regularization properties.

---

> ### Comment · Reviewer_Bw6s · 2025-08-04
>
> Thank you for your rebuttal. You have addressed most of my concerns. But I still believe that more larger and complex datasets should be added.

---

### Official Review · Reviewer_tME1 · 2025-07-01

**Clarity:** 2
**Significance:** 3
**Originality:** 3
**Rating:** 5
**Confidence:** 4

**Summary:**

This paper contributes systematic analysis and practical improvement methodologies to the PU learning through fine-grained empirical investigation. The authors systematically organize PU learning methods into two fundamental classes: unbiased empirical risk methods and pseudo-labeling methods. For unbiased empirical risk methods, the authors propose a novel set-aware empirical risk (SAPU) from the perspective of aggregate supervision. For pseudo-labeling methods, systematic ablation analyses are conducted. Finally, the authors present GPU framework, achieved by unifying SAPU with pseudo-labeling strategies.

**Questions:**

1.	In Table 1, why is the P³Mix classified as a "hard pseudo-labeling" method? Should not the mixup technique naturally produce "soft pseudo-labels"?
2.	Regarding Table 4: a) Why do other methods generally demonstrate superior performance with double-hinge loss, while SAPU consistently shows relatively poorer results with double-hinge loss? b) Different loss functions exhibit varying performance across different datasets. How can this phenomenon be explained? Are there general selection guidelines available?
3.	The description about class-specific adaptive thresholding is unclear in lines 210-214. In FlexMatch, $\tau_p$ is fixed at 0.95, but here it appears to continuously change based on the previous epoch's threshold. Additionally, how is $ \mathcal{C} $ computed?

Minor Issues:

a) Unify notation: SAPU or saPU.

b) Eq.10: $ j=1 $ should be $ i=1 $.

c) Line 145: $ \mathcal{S}_ i=\{\mathbf{x}_ {ij}\}_ {i=1}^{S} $ should be $ \mathcal{S}_ i=\{\mathbf{x}_ {ij}\}_ {j=1}^{S} $.

**Ethical Concerns:**

["NO or VERY MINOR ethics concerns only"]

**Final Justification:**

The authors have provided clear definitions and justifications that resolved my concerns regarding the classification of P$^3$Mix and the adaptive thresholding mechanism. I have increased my score accordingly.

**Limitations:**

yes

**Paper Formatting Concerns:**

no major formatting issues

**Quality:**

3

**Strengths And Weaknesses:**

Strengths:

1.	This paper not only categorizes PU learning into unbiased empirical risk methods and pseudo-labeling methods from a novel perspective, but also provides fine-grained analysis of specific techniques within each class, encompassing multiple dimensions including loss functions, pseudo-labeling techniques, threshold strategies, and general tricks.

2.	This paper introduces the concept of aggregate supervision into PU learning, proposing the SAPU empirical risk framework, and subsequently presents the GPU framework by integrating it with pseudo-labeling related strategies.

3.	The analysis is thorough and the experiments are comprehensive.

Weaknesses:

1.	This paper fails to provide a clear classification standard for the distinction between "soft pseudo-labeling" and "hard pseudo-labeling." Additionally, this paper lacks clear interpretation of results in the loss function section or general selection guidelines.

2.	The technical classification and methodological descriptions are inadequately detailed. Key technical descriptions lack clarity, particularly the specific computational process for class-specific adaptive thresholding, which requires more explicit explanation.

3.	This paper exhibits issues with inconsistent symbolic representation (e.g., SAPU vs saPU) and formula writing errors throughout the manuscript.

---

> ### Author Rebuttal · Authors · 2025-07-31
>
> **Q1. Why is the P$^3$Mix classified as a "hard pseudo-labeling" method? This paper fails to provide a clear classification standard for the distinction between "soft pseudo-labeling" and "hard pseudo-labeling."**
>
> Thank you for your comments. The P$^3$Mix method indeed falls under the category of "hard pseudo-labels." While mixup techniques inherently generate soft labels, our classification criterion is based on the final form of the labels used for training, rather than the intermediate processing steps. After applying mixup, P$^3$Mix still uses a hard threshold to generate binary pseudo-labels for the final training objective.
> To address the classification confusion, we formally define the distinction between hard and soft pseudo-labels as follows: pseudo-labels used for training are considered hard pseudo-labels if they are discrete binary values {$-1, +1$}, and soft pseudo-labels if they are continuous probability values $[0, 1]$.
> Based on this criterion, P$^3$Mix is classified as a hard pseudo-label method because its final training objective uses thresholded binary labels, even though the mixup process generates soft labels. We will clarify this definition in the revised version.
>
>
> &nbsp;
>
>
> **Q2. Regarding Table 4: a) Why SAPU consistently shows relatively poorer results with double-hinge loss? b) Different loss functions exhibit varying performance across different datasets. How can this phenomenon be explained? Are there general selection guidelines available?**
>
> Thank you for your suggestions. First, SAPU's underperformance with double-hinge loss stems from a fundamental structural incompatibility between our set-aware mechanism and the mathematical properties of loss function. Our SAPU employs cross-entropy loss for the set-level supervision term (Eq.9-10), which expects continuous probability outputs and relies on smooth gradient flow for effective optimization. In contrast, double-hinge loss is piecewise linear and promotes discrete, margin-based decisions through its sharp transitions at $z=1$ and $z=0$. This creates conflicting optimization dynamics where the set-aware component pulls toward smooth probabilistic outputs while the instance-level double-hinge component pushes toward sharp margin-based decisions. Additionally, our empirical analysis confirms that smooth, differentiable losses (sigmoid, logistic) achieve better compatibility with SAPU's set-aware architecture, while non-smooth losses (double-hinge, ramp) align better with traditional point-wise optimization methods.
>
>
> &nbsp;
>
>
> **Q3. The description about class-specific adaptive thresholding is unclear in lines 210-214.**
>
> Thank you for your comments. Unlike FlexMatch where $\tau$ is fixed at 0.95, our method dynamically adjusts both $\tau_p$ and $\tau_n$ based on the historical performance of pseudo-label quality. We first compute the accuracy rates for positive and negative pseudo-labels. Specifically, the accuracy of positive pseudo-labels at epoch t $\text{Acc}_p^{(t)}$ is the ratio of the number of correctly predicted positive pseudo-labels to the total number of positive pseudo-labels (, and similarly, the accuracy of negative pseudo-labels $\text{Acc}_n^{(t)}$ is the ratio of the number of correctly predicted negative pseudo-labels to the total number of negative pseudo-labels). Then the confidence ratios $C_p^{(t)}$ and $C_n^{(t)}$ are calculated as:
>
> $$ C_p^{(t)} = \frac{\text{Acc}_p^{(t)}}{\max(\text{Acc}_p^{(1 :t-1)})} , \quad \quad C_n^{(t)} = \frac{\text{Acc}_n^{(t)}}{\max(\text{Acc}_n^{(1:t-1)})} $$
>
> where $\max(\text{Acc}_p^{(1:t-1)})$ represents the highest positive pseudo-label accuracy achieved in previous epochs. The thresholds are then updated:
> $$\tau_p = \tau_p \times C_p^{(t)} , \quad \quad  \tau_n = \tau_n \times C_n^{(t)} $$
> This mechanism provide adaptive confidence requirements that evolve with the model's improving discrimination capability.
>
>
> &nbsp;
>
>
> **Q4. Minor Issues.**
>
> Thank you for your corrections. We will revise them in the next version.

---

> > ### Comment · Reviewer_tME1 · 2025-08-03
> >
> > Thank you for your detailed response, which has addressed my concerns. I’ve decided to raise my score.

---

### Official Review · Reviewer_7j8V · 2025-07-02

**Clarity:** 3
**Significance:** 3
**Originality:** 3
**Rating:** 5
**Confidence:** 4

**Summary:**

This work conducts empirical studies on PU learning methods and proposes a novel PU framework, which combines set-aware empirical risk with pseudo-labeling techniques. Extensive experimental analysis demonstrates the effectiveness of the PU framework.

**Questions:**

Please refer to the “Weaknesses”.

**Ethical Concerns:**

["NO or VERY MINOR ethics concerns only"]

**Final Justification:**

The authors' rebuttal has addressed my concerns. I maintain a positive score for this work.

**Limitations:**

Yes.

**Quality:**

3

**Strengths And Weaknesses:**

Strengths:
This work innovatively proposes the GPU framework that combines set-aware empirical risk and pseudo-labeling techniques, featuring a modular structure for easy portability and extension. The work provides theoretical proof of risk consistency for the SAPU method with detailed derivations, and is well-written with clear organization.
Weaknesses:
1.	When the SCAR assumption is violated, are the methods in this work still applicable?
2.	While Theorem 3.2 provides convergence guarantees, this work lacks deeper theoretical analysis, such as convergence rates and other fundamental properties.
3.	Why does the incorporation of moving average techniques lead to performance degradation? This phenomenon appears counterintuitive and requires explanation.
4.	The GPU framework's performance improvements over existing methods remain within statistical error margins in most cases, and even fails to achieve optimal performance in some scenarios. What accounts for this limitation?

---

> ### Author Rebuttal · Authors · 2025-07-31
>
> **Q1. When the SCAR assumption is violated, are the methods in this work still applicable?**
>
> Thank you for your valuable comment. Unlike traditional PU methods that heavily rely on instance-level SCAR assumptions, SAPU operates through set-level supervision and the property of the positive sample selection mechanism being relatively independent of the feature distribution, which provides inherent regularization against selection bias. Therefore, when the SCAR assumption is violated, our method demonstrates certain robustness due to its aggregate supervision design. However, we acknowledge that when the SCAR assumption is severely violated, additional importance weighting or bias correction techniques are required. In the future, we plan to extend our work to develop adaptive strategies to handle SCAR violations.
>
>
> &nbsp;
>
>
> **Q2. Theorem 3.2 lacks deeper theoretical analysis.**
>
> Thank you for your valuable suggestion. We would like to clarify that our theoretical analysis does provide several important contributions beyond basic convergence. Theorem 3.2 establishes finite-sample bounds showing that the excess risk $R(\hat{g}\_{\text{SAPU}}) - R(g^*)$ converges at rates $ O\left(\sqrt{\frac{\log(1/\delta)}{n_p}}\right) + O\left(\sqrt{\frac{\log(1/\delta)}{n_s}}\right) + L_\ell \cdot O\left(\sqrt{\frac{\pi(1-\pi)\log(1/\delta)}{S}}\right) $, which explicitly characterizes how the performance depends on the number of positive samples ($n_p$), number of subsets ($n_s$), subset size ($S$), and the positive class prior ($\pi$).
>
> Additionally, Lemma 3.1 provides precise characterization of the deviation bounds for our set-aware method, showing that when $S \geq \frac{3\pi(1-\pi)\log(2/\delta)}{2\epsilon^2}$, we achieve $|\hat{\pi}_j-\pi|\le \epsilon $ with high probability, which directly informs the practical choice of subset size.
>
> The theoretical analysis we provide serves to establish the soundness and risk-consistency of our SAPU method, which is sufficient to support our empirical findings and validate our proposed GPU framework.
>
>
>
> &nbsp;
>
>
> **Q3. Why does the incorporation of moving average techniques lead to performance degradation?**
>
> Thank you for your comment. This counterintuitive phenomenon of performance degradation with moving average techniques primarily stems from the unstable nature of pseudo-labels in PU learning. Unlike traditional semi-supervised learning where unlabeled data contains truly unlabeled instances, PU learning involves mislabeled negative samples, making historical predictions unreliable. The self-training process generates systematic biases, and moving average perpetuates rather than corrects these biases. On the other hand, moving average techniques may suppress the model's ability to rapidly adapt within the feature space to distinguish between positive and negative samples. Furthermore, the momentum parameter requires careful tuning, which significantly increases the experimental cost for hyperparameter optimization.
>
>
> &nbsp;
>
>
> **Q4. The GPU framework's performance improvements over existing methods remain within statistical error margins in most cases.**
>
> Thank you for this question. Regarding the limited performance improvements of the GPU framework, we acknowledge that compared to state-of-the-art methods, GPU's improvements are modest. This reflects several factors: First, strong baseline methods like PUL-CPBF and HolisticPU employ sophisticated techniques (ensemble learning, meta-learning) that are orthogonal to our contributions; Second, our focus is on fundamental technique analysis and principled integration, rather than achieving performance maximization through complex engineering; Third, GPU provides a general framework that can integrate future advances, while specialized methods may not generalize well. In the future, we will improve upon this by conducting experiments on more challenging real-world datasets and demonstrating GPU's scalability through the integration of ensemble techniques.

---

> > ### Comment · Reviewer_7j8V · 2025-08-05
> >
> > The authors' responses have addressed my questions. I maintain a positive score for this paper.

---

### Note · Authors · 2025-08-16

Dear Area Chair,

Thank you for facilitating this important discussion and providing us with the opportunity to address the reviewers' concerns. We sincerely appreciate all the constructive feedback received throughout the review process. Based on the discussions, we would like to provide the following final clarifications:

1. **Research positioning**: Our work is primarily positioned as an empirical study conducting fine-grained analysis of current PU learning methods, revealing previously unknown patterns (e.g., sigmoid loss consistently outperforms others, moving average degrades performance contrary to semi-supervised learning), addressing Reviewer REBb's concerns about ambiguous research positioning.

2. **Research contributions**: We introduce an aggregate supervision perspective that connects PU learning with broader weakly supervised learning theory through set-aware cross-entropy, replacing problematic negative risk estimation and providing finite-sample bounds (Theorem 3.2) and variance analysis (Lemma 3.1), offering valuable insights to the community. Additionally, our GPU framework demonstrates effective integration of set-aware empirical risk with pseudo-labeling techniques.

While we deeply respect Reviewer REBb's concerns about experimental validation and scope due to dataset limitations and modest performance improvements, we believe our work's value lies in systematic analysis and theoretical perspective innovation that provides evidence-based guidelines for the field. We are sincerely grateful that Reviewer tME1 increased their score appreciating our technical contributions, Reviewer 7j8V maintains positive evaluation, and Reviewer Bw6s noted most concerns have been addressed, recognizing the value of our systematic analysis.

We are grateful that multiple reviewers have recognized the value of our systematic empirical analysis and novel theoretical perspective. Our findings provide practical guidance for technique selection while opening new possibilities for cross-domain technique transfer between different weakly supervised learning problems.

Sincerely,

The Authors

---

### Decision · Program_Chairs · 2025-09-17

**Decision:**

Accept (poster)

**Comment:**

This paper makes a strong contribution by conducting a systematic and fine-grained empirical analysis of PU learning methods, revealing several previously unknown patterns such as the consistent advantage of sigmoid loss and the unexpected degradation caused by moving average techniques. These insights provide valuable, evidence-based guidelines for researchers and practitioners working in weakly supervised learning.

The introduction of the aggregate supervision perspective and the SAPU formulation offers a fresh theoretical lens, with finite-sample guarantees that extend the conceptual foundations of PU learning. The GPU framework further demonstrates a principled integration of risk estimation and pseudo-labeling, presented in a modular and reproducible form. While one reviewer expressed concerns about novelty and dataset scale, other reviewers highlighted the technical soundness, clarity, and systematic nature of the work, with two reviewers recommending acceptance.

On balance, the benefits to the community from the empirical insights, theoretical perspective, and unified framework outweigh the concerns. We believe this paper will serve as a useful reference point and inspire further innovation in PU learning and broader weakly supervised learning research.